# Serum Concentrations of Thyroid-Stimulating Hormone, Triiodothyronine, and Thyroxine in Outpatients Infected with SARS-CoV2 in Khuzestan Province, Iran: A Disease Clinical Course Approach

**DOI:** 10.3390/medicina58070891

**Published:** 2022-07-02

**Authors:** Mahshid Naghashpour, Ali Darvishi, Maryam Adelipour, Reza Bagheri, Alexei Wong, Katsuhiko Suzuki, Sahar Golabi

**Affiliations:** 1Department of Nutrition, School of Medicine, Abadan University of Medical Sciences, Abadan 63138-33177, Iran; m.naghashpour@abadanums.ac.ir; 2School of Medicine, Abadan University of Medical Sciences, Abadan 63138-33177, Iran; adarvishiy@yahoo.com; 3Department of Biochemistry, School of Medical Sciences, Ahvaz Jundishapur University of Medical Sciences, Ahvaz 61357-15794, Iran; adelipour-m@ajums.ac.ir; 4Department of Exercise Physiology, University of Isfahan, Isfahan 81746-73441, Iran; will.fivb@yahoo.com; 5Department of Health and Human Performance, Marymount University, Arlington, VA 22207, USA; alexei.wong@marymount.edu; 6Faculty of Sport Sciences, Waseda University, 2-579-15 Mikajima, Tokorozawa 359-1192, Japan; katsu.suzu@waseda.jp; 7Department of Medical Physiology, School of Medicine, Abadan University of Medical Sciences, Abadan 63138-33177, Iran

**Keywords:** clinical symptoms, COVID-19, SARS-CoV2, T3, T4, TSH

## Abstract

*Background and Objectives:* The virus SARS-CoV2, which causes COVID-19, affects the endocrine system. This study investigated serum concentrations of the thyroid-stimulating hormone (TSH), triiodothyronine (T3), and thyroxine (T4) in 53 outpatients infected with SARS-CoV2 and 53 non-infected matched participants in Khuzestan Province, Iran. We also examined the possible association of clinical symptoms progression and disease severity with serum concentrations of TSH, T3, and T4. *Materials and Methods:* A checklist was applied to collect demographic and clinical data. Blood samples were taken for biochemical analysis of serum concentrations of TSH, T3, and T4. Clinical symptoms of the infected outpatients were monitored weekly for 28 days. *Results:* Our results indicated that, as the severity of the disease increased, the respiratory and pulse rates raised significantly. Additionally, disease severity was significantly different between genders. Specifically, 79.5% of the asymptomatic/mild, and 38.5% of moderate outpatients were men. We also found significantly lower serum T3 but higher T4 in infected outpatients, compared with controls. However, serum TSH did not significantly differ between the two groups. The generalized estimating equation (GEE) analysis revealed no relationship between clinical symptoms progression and disease severity with serum concentrations of TSH, T3, and T4 in our study population. Additionally, GEE analysis showed that the odds ratio of neurological symptoms among women was 2.5 times that of men, the odds ratio of neurological symptoms in illiterates was 10 times higher than that of those without a high-school diploma, and the chance of developing pulmonary symptoms in those without high-school diploma was about 21 times higher than illiterates. *Conclusion:* In conclusion, this study showed that infected outpatients had significantly lower serum T3 but higher T4 than non-infected participants. There was no relation between symptom progression and disease severity with serum concentrations of TSH, T3, and T4, but educational status and sex significantly affected the chance of neurological and pulmonary symptoms occurring over 28 days. Our results may be used to develop potential therapies to treat COVID-19 disease.

## 1. Introduction

Severe acute respiratory syndrome 2 (SARS-CoV2) is responsible for the coronavirus 2019 (COVID-19) disease, which became a pandemic in 2020 [1]. Some endocrine organs, including the thyroid, have been shown to secrete the angiotensin-converting enzyme 2 (ACE2), the SARS-CoV2 receptor, for entry into the cell. Although the interaction between ACE2 and SARS-CoV2 is expected to elicit an endocrine response, there is limited clinical or preclinical evidence to support this. The thyroid is an essential endocrine gland, as it influences all aspects of metabolism, consequently having a massive impact on overall health. However, there is very little information available about the effects of coronaviruses on the thyroid gland. A 2003 study on the prevalence of SARS found that serum concentrations of triiodothyronine (T3) and thyroxine (T4) in patients with SARS were lower than in acute and congestive control groups [2]. Moreover, an autopsy study of five patients with SARS showed thyroid gland damage, especially of follicular and parafollicular cells. The destruction of follicular cells indicates low serum concentrations of T3 and T4 [2]. Additionally, Chen et al. reported lower total serum concentrations of triiodothyronine and thyroid-stimulating hormone (TSH) in 50 Chinese patients with a definitive diagnosis of COVID-19 when compared with control participants [2]. These concentrations returned to normal levels following COVID-19 recovery, leading researchers to conclude that changes in serum concentrations of TSH and triiodothyronine might be important manifestations of the disease. In a study by Wang et al., serum concentrations of T3 and TSH were lower in Chinese patients with COVID-19 than in controls. Among the COVID-19 patients, 69% had thyroid dysfunction. Additionally, it was found that the ratio of patients with severe to moderately severe conditions was related to thyroid dysfunction [3].

Although previous investigations have shown a relationship between the severity of various diseases and endocrine function, there are contradictory results regarding the function of the pituitary–thyroid axis in patients of various COVID-19 severities. Moreover, there is a lack of clinical evaluations of TSH, T3, and T4 concentrations in patients with COVID-19 of different ethnicities. As factors such as race and geographical area influence TSH, T3, and T4 concentrations and COVID-19 severity, it is essential to evaluate the concentrations of these hormones and their relationship to disease severity and progression in different populations with COVID-19 [4,5]. Consequently, the purpose of this study was to evaluate serum concentrations of TSH, T3, and T4 and their relationship to disease severity and progression in the outpatients infected with SARS-CoV2 in the southwest region of Khuzestan Province, Iran, with the approach of disease clinical course. We also compared serum concentrations of these factors between outpatients with different COVID-19 severities and between COVID-19 outpatients and matched controls.

## 2. Materials and Methods

### 2.1. Participants

The study population consisted of individuals who had clear (positive or negative) polymerase chain reaction (PCR) results for COVID-19, which were divided into two groups: outpatients with COVID-19 (positive-PCR) and non-infected controls (negative-PCR). The inclusion criteria were ≥11 years of age, male and female gender, a straightforward (positive, negative) PCR result, willingness to participate in the study, understanding the relevant information, and completing the informed consent form. Exclusion criteria were pregnancy and lactation; uncertain PCR test results; PCR testing for a second time; smoking; underlying thyroid disease; being on medication for a thyroid disorder and/or failure to assess thyroid function; occupation in high-risk jobs, including health staff and public transport drivers (for the control group); and any circumstances that the researcher did not consider as justifying the participation of individuals in the study.

### 2.2. Study Design

This was a health-service-center-based cross-sectional and descriptive-analytical study. We aimed to compare demographics, baseline comorbidities, and serum concentrations of TSH, T3, and T4 in COVID-19-infected outpatients and non-infected participants. The participants were divided into two groups—negative (non-infected participants) and positive (infected outpatients) PCR (N = 53 in each group). Outpatients enrolled in the positive-PCR group were divided into two categories based on reported clinical signs—namely, asymptomatic or pre-symptomatic/mild and moderate [6]. Infected outpatients were followed from day 1 to day 28 to evaluate the effect of baseline serum concentrations of TSH, T3, and T4 on the symptoms progress of COVID-19. Participants’ demographics, baseline comorbidities, and clinical data were recorded on day 1 at the Imam Khomeini Health Center in Abadan city, Iran, between June 2020 and August 2020. The assessment of demographics comprised the following items: age, gender, tobacco consumption status, marital status, education status, duration of the infection, history of blood transfusions or plasma products, hospitalization due to respiratory problems, history of contact with COVID-19 suspect persons, the time interval between encountering a COVID-19 suspicious person and visiting the health center, and travel history in the last two months. Clinical measurements included respiratory rate, pulse rate, oxygen saturation (SpO_2_), and blood sampling. Serum concentrations of TSH, T3, and T4 were measured using an IDEAL kit (IDEAL, Tehran, Iran) following the ELISA method. The researcher gathered information on the patients’ clinical symptoms (from positive-PCR group outpatients) via phone on the 7th, 14th, 21st, and 28th days [7,8]. All symptoms fell into four categories: general, pulmonary, gastrointestinal, and neurologic symptoms [9]. All testing procedures were approved by the Ethics Committee of the Abadan University of Medical Sciences (Ethics Code: IR.ABADANUMS.REC.1399.102).

### 2.3. Clinical Symptoms Evaluation

As previously described, we divided COVID-19 symptoms into four categories: general, pulmonary, gastrointestinal, and neurologic. General symptoms included fatigue, fever, night sweating, shivering, low body temperature, runny nose, and sore throat. Chest pain, cough, and shortness of breath were pulmonary symptoms. The gastrointestinal symptoms included nausea, vomiting, anorexia, diarrhea, constipation, blowing, and stomachache, while the neurologic one consisted of muscle pain, ear pain, joint pain, taste/odor disorder, and headache [9].

### 2.4. Evaluation of Comorbidities

Using a researcher-made questionnaire, the presence of the underlying diseases, including high blood pressure, diabetes, obesity, malnutrition, cancer, liver disease, chronic pulmonary disease, chronic neurological disease, chronic hematologic disease, chronic renal disease, chronic heart disease, Acquired immunodeficiency syndrome/human immunodeficiency virus (AIDS/HIV), dementia, rheumatoid arthritis, asthma/allergy, intestinal ulcer, underlying thyroid disease, and unclear thyroid function were assessed in an outpatient self-report manner.

### 2.5. Statistical Analysis

An expert in statistics determined that the minimum number of participants required to enter the study was at least 38 patients with SARS-CoV2 and 38 healthy individuals (as controls). This was based on TSH as a factor in a prior study [3]. The following formula was used for the calculation:n = (〖(σ〗_1^2 + σ_2^2) 〖(z_(1-α/2) +z_(1-β)) 〗^2)/δ^2
where δ^2 = (μ_1-μ_2)^2 for which there was a significance level of 5% (α = 5%), test power of 80% (β = 2.0), mean (μ_1 = 0.62) with standard deviation (σ_1 = 0.62), and mean (μ_2 = 1.10) with standard deviation (σ_2 = 0.84). Taking into account the probabilities and losses, finally, 53 individuals in each group were examined. The study samples were selected among the statistical population that met the inclusion criteria, following a simple random sampling method. We matched the data of the infected outpatients with those of the non-infected individuals of the same sex and age. Statistical analysis was performed using SPSS software (SPSS Inc. Chicago, IL, USA) version 21. The normality of the data was checked with the Kolmogorov–Smirnov test. The results are displayed as mean ± SD if the data were normal. Otherwise, the middle and the confidence interval values (95% CI) are provided. Data were compared between the groups by analysis of variance (ANOVA) and independent sample *t*-test for continuous variables. The significance level was considered 0.05. Based on the longitudinal data that display repeated outcomes within one individual, the generalized estimating equation (GEE) technique was used for analysis. The GEE model with AR (1) structure correlation was used to analyze a longitudinal dataset with 6 measurements on a positive-PCR group (53 participants) for each of the 4 dichotomous outcome variables (general, pulmonary, gastrointestinal, and neurologic symptoms) separately. The odds ratio (OR) and the confidence interval for OR are reported for each model.

## 3. Results

### 3.1. Demographics, Baseline Comorbidities, and Clinical Symptoms Assessments

General characteristics, comorbidities, and clinical presentations of participants are presented in Table 1. Our results revealed that for negative, asymptomatic/mild positive, and moderate positive-PCR groups, increased disease severity led to significant increases in mean respiratory rate (13 ± 1.9, 13.5 ± 1.1, and 15.9 ± 1.7, respectively, *p* ≤ 0.001) and mean pulse rate (87.2 ± 13.4, 87.3 ± 17, 100 ± 18, respectively, *p* = 0.023).

Our results also showed a significant difference between males and females regarding disease severity. Specifically, 79.5% of the asymptomatic/mild positive-PCR group were men, while 20.5% were women (*p* = 0.02). Moreover, 38.5% of the moderate positive-PCR group were men, and 61.5% were women (*p* = 0.02). Given that the data analysis did not show a significant difference between negative- and positive-PCR groups in terms of gender, it is clear that the prevalence of moderate severity was higher in women. In contrast, asymptomatic/mild severity prevalence was higher in men.

Regarding comorbidities, we found that the prevalence of asthma/allergy was higher in the moderate positive-PCR group compared with the asymptomatic/mild positive-PCR group (*p* ≤ 0.001). No significant differences in comorbidities were observed between positive and negative-PCR groups and between different severities of COVID-19 in the positive-PCR group.

### 3.2. The Ratio of the Chances of Occurrence of Different Types of COVID-19 Symptoms over Time

We assessed the presence and types of possible associations between estimated parameters (age, sex; marital status; education status; serum concentrations of TSH, T3, and T4, and BMI) with the chances of different types of COVID-19 symptoms occurring over time in the positive-PCR group. The results indicated that baseline serum concentrations of TSH, T3, and T4; age; marital status; and BMI did not affect the chances of experiencing any of the four groups of COVID-19 symptoms over time.

We also found that educational status and sex significantly affected the chance of neurological symptoms occurring over 28 days. Specifically, the odds ratio of the occurrence of neurological symptoms among positive-PCR women during the 28-day follow-up period was 2.5 times that of positive-PCR men (OR = 0.4 (0.2–0.8), *p* = 0.007). Our findings also revealed that the odds ratio of neurological symptoms in illiterate individuals during the 28-day study period was 10 times higher than in individuals without a high-school diploma education (OR = 0.1 (0.03–0.7), *p* = 0.017). Moreover, it was revealed that educational status significantly affected the chance of pulmonary symptoms occurring over 28 days. Specifically, the chance of developing these symptoms in outpatients without a high-school diploma was about 21 times higher than in illiterates (OR = 20.9 (1.2–350.9), *p* = 0.03). All results are summarized in Table 2.

### 3.3. Laboratory Evaluation of Serum Concentrations of TSH, T3, and T4 and Possible Relationship with the Clinical Course of the Disease

A comparison of serum concentrations of TSH, T3, and T4 among different clinical classifications of COVID-19 according to its severity is shown in Figure 1A–C. Our results revealed that the mean serum T3 is significantly lower in the asymptomatic/mild (1.5 ± 0.2) and moderate (1.6 ± 0.2) positive-PCR groups, compared with the negative (1.7 ± 0.3)-PCR group (*p* = 0.000 and *p* = 0.018, respectively) (Figure 1A). An analysis of data regarding serum T4 showed significantly higher serum T4 in asymptomatic/mild (10.8 ± 2.4) and moderate (10.5 ± 2.4) positive-PCR groups, compared with the negative (8.5 ± 1.7)-PCR group (*p* = 0.000 and *p* = 0.002, respectively) (Figure 1B). In terms of serum TSH, there was no significant difference between asymptomatic/mild (3.8 ± 4.7) and moderate (2.3 ± 1.5) positive-PCR groups, compared with the negative (3.4 ± 3.3)-PCR group (Figure 1C).

Overall, our results showed significant differences between the negative- and positive-PCR groups regarding their serum concentrations of T3 and T4 (*p* ≤ 0.001 for both) but not their serum TSH. However, no significant differences were found between the asymptomatic/mild and moderate positive-PCR groups regarding their serum concentrations of TSH, T3, and T4 (*p* ˃ 0.05).

## 4. Discussion

Our findings indicate that infected outpatients had significantly lower serum T3 but higher T4 than non-infected participants. However, no relationship was found between clinical symptoms progression and disease severity with serum concentrations of TSH, T3, and T4. Moreover, outpatient respiratory and pulse rates raised concurrently with COVID-19 severity. We also revealed that disease severity was significantly different between genders, and asthma/allergy as comorbidity increased with increasing severity of the disease. Additionally, educational status and sex significantly affected the chance of neurological and pulmonary symptoms occurring over 28 days.

Even though considerably discussed regarding the clinical course, prognostic inflammatory markers, and disease complications of COVID-19, the possible interaction between SARS-CoV2 and the thyroid is complicated and inadequately comprehended. In contrast to SARS-CoV1, the influence of SARS-CoV2 on thyroid function is debatable. Chen et al. compared the clinical characteristics of Chinese patients with COVID-19 in a deceased group and a recovery group and reported that serum concentrations of TSH and free T3 (FT3) were significantly less in the deceased patients than in the recovered patients. The difference in free T4 (FT4) was not significant [2]. Another study analyzed thyroid function between Chinese COVID-19 patients and healthy control participants [10]. During the follow-up period of three months after the diagnosis of COVID-19, 64% of the patients had abnormal thyroid performance. Of these patients, 56% had lower than normal TSH, which was higher than the healthy control group. Additionally, TSH and total T3 (TT3) concentrations in patients with COVID-19 were significantly lower than in the healthy control group. Nevertheless, no significant difference in total T4 (TT4) was detected. Furthermore, in a subgroup analysis of COVID-19 patients, the decrease in serum concentrations of TSH and TT3 positively correlated with the severity of the disease. The more severe the COVID-19 infection was, the lower the serum concentrations of TSH and TT3 were. Similar results were also reported in two other studies that included Chinese and Italian cohorts [11,12].

A Recent meta-analysis by Giovanella et al. found that the relationship between the thyroid gland function and the COVID-19 is complicated and controversial. Most COVID-19 patients were euthyroid with serum TSH within the normal 44–94% range. These findings align with our results, indicating no significant difference between serum TSH in the negative- and positive-PCR groups. Thyroid dysfunction in COVID-19 patients varies significantly among the included studies, from 13% to 64%. Clinical presentations of COVID-19 vary significantly as well. Moreover, a positive correlation between thyroid dysfunction and clinical severity of COVID-19 was reported [13].

Our results also showed that increased disease severity led to significant increases in mean respiratory and mean pulse rates. One of the most commonly used clinical screening tools to identify lower respiratory tract infections is the number of breaths per minute. As long as COVID-19 harms the respiratory system, it is rational to believe that alterations in respiratory efficiency and, therefore, resting respiratory rate might occur during the early stages of infection [14]. In line with the results of our study, Zhou et al. classified young Chinese adults diagnosed with COVID-19 as having a mild or severe disease based on their respiratory rate [15], which confirmed the validity of the classification technique [15].

Coronaviruses are also known to affect the cardiovascular system [16]. Natarajan et al. assessed physiological signs associated with COVID-19 in 2745 participants diagnosed with COVID-19 (active infection, PCR test) [17]. The authors reported that the illness usually raises respiration and heart rates [17], which agrees with our outcomes. Miller et al. investigated whether alterations in respiratory rate could serve as a leading indicator of SARS-CoV2 infections. They developed a model to approximate the probability of SARS-CoV2 infection based on changes in respiratory rate during nighttime sleep. Their model is able to differentiate between healthy and infected days for those who confirmed positive for COVID-19 and those who had symptoms but tested negative. Furthermore, the model recognized 20% of COVID-19 positive individuals in the validation dataset in the two days prior to symptom onset and 80% of COVID-19 positive cases by the third day of symptoms. Consequently, researchers suggest that the early stages of the infection may have a noticeable signature according to respiratory rate, which aligns with our findings [14].

In another study, Chinese patients suspected of having COVID-19 were investigated. The investigators prospectively analyzed data from patients with laboratory-confirmed COVID-19 infection via RT-PCR and next-generation sequencing. They found that respiratory rate is significantly higher in ICU care patients than in non-ICU care patients, which revealed a relationship between respiratory rate and disease severity [18], which agrees with our findings. The authors also showed a direct and significant relationship between disease severity and respiratory rate [18].

We also found that the prevalence of asthma/allergy comorbidity was higher in the moderate positive-PCR group compared with the asymptomatic/mild positive-PCR group, indicating a direct relationship between asthma/allergy prevalence and disease severity. Asthma is one of the most widespread, long-lasting diseases in the United States (about 8–9% of the population). Respiratory viruses are recognized as triggers of asthma exacerbations. Coronaviruses are respiratory viruses and have been implicated in both upper respiratory tract infections and asthma exacerbations. Studies suggest that the prevalence of asthma in the COVID-19 population in China was <1%. In the meantime, the described prevalence of asthma in patients with COVID-19 in the United States differs from 7.4% to 17% [19], which indicates the magnitude of assessing the comorbidities of COVID-19 patients of different races and ethnicities.

Presently, the CDC categorizes patients with underlying moderate-to-severe asthma as a high-risk group vulnerable to severe COVID-19, which supports our outcomes. The Centers for Disease Control and Prevention states that patients with moderate-to-severe asthma belong to a high-risk group vulnerable to severe COVID-19. Nevertheless, the association between asthma and COVID-19 has not been well-known [19]. A previous investigation in a major US health system indicated that the occurrence of asthma among patients with COVID was 14%, which is close to the value found in our study (11.1%). Nevertheless, asthma was not related to an augmented risk of hospitalization following adjusting for age, sex, and other comorbidities [19]. Surprisingly, the incidence of asthma (0.9%) in patients with COVID-19 was evidently less than that in the adult population of Wuhan (6.4%). Henceforth, the investigators considered that there might be a TH2-mediated decreased vulnerability to COVID-19 in patients with asthma [20]. A new review study including an additional 12 predominantly Chinese COVID-19 cohorts/cases (874 patients) indicated that asthma was “surprisingly underreported” [21] and absent in a Chinese nationwide analysis of 1590 COVID-19 cases. Postulated reasons included a lack of chronic airways disease awareness and community spirometry testing [22]. On the other hand, a more recent case series from New York of 393 consecutive established COVID-19 admissions documented an asthma rate of 12.5%, slightly higher than the occurrence of current adult asthma of 10.1% in New York State [23]. Additionally, an earlier meta-analysis of 744 asthmatic patients and 8151 non-asthmatic patients showed that the occurrence of asthma had no meaningful influence on mortality from COVID-19. In addition, a descriptive examination of further clinical outcomes showed no difference in the length of hospitalization or the risk of ICU transfer between asthmatic and non-asthmatic patients [2]. Although we found the prevalence of asthma comorbidity is higher in the moderate positive outpatients, preliminary data indicates that asthma (as comorbidity) may not increase mortality from COVID-19. Further studies should evaluate the influence of asthma on COVID-19 disease prognosis to draw firm conclusions.

We found that the odds ratio of neurological symptoms was higher in women than in men, while the odds ratio of neurological symptoms in illiterates was higher than that of patients without a high-school diploma. Furthermore, the chance of developing pulmonary symptoms in those without a high-school diploma was higher than in illiterates. Sex and education level are significant epidemiological factors for several diseases’ incidence, severity, prognosis, and outcome. Falagas et al. assessed sex differences in the occurrence and severity of respiratory tract infections and showed that females were more usually influenced by infections of the upper respiratory tract; however, males were more commonly influenced by infections of the lower respiratory tract [24]. In line with our results, another study revealed that the odds ratio of neurologic symptoms of COVID-19 in males was 0.41 times that in females in the same follow-up period [7]. In line with our findings regarding sex effect on severity and incidence of COVID-19 symptoms, Alwani et al. investigated sex-based differences in severity and mortality in COVID-19. They reported that SARS-CoV2 has a male bias in severity and mortality. Their report is consistent with previous coronavirus pandemics such as SARS- CoV and MERS-CoV, and viral infections in general [25]. Zavaleta et al. investigated the effect of the level of education on mortality of SARS-CoV2. They revealed that social factors, such as lower educational acquisition, are related to the morbidity and mortality of COVID-19. They concluded that low levels of education are related to high incidence and prevalence of cardiovascular and cerebrovascular diseases, cancer, diabetes, hypertension, and chronic respiratory disease, and patients with these diseases are at increased risk of severe disease due to COVID-19 because of low levels of immune cells and high levels of cytokines in body fluids [26]. A UK-based study also investigated the role of ethnicity and socioeconomic position in the development of SARS CoV2 infection. It has been obtained that socioeconomic deprivation and low educational achievement were constantly related to a high risk of proven infection [27]. The results reported by both Zavaleta et al. and Niedzwiedz et al. are in line with our results regarding the effect of education level on the prognosis of COVID-19. In a short report, Regidor et al. assessed educational level (third level, upper secondary, lower secondary, and elementary or less) and mortality from infectious diseases (including tuberculosis, bacterial meningitis, septicemia, AIDS, hepatobiliary disease, mycoses, some selected respiratory tract infections, etc.). Following adjusting for all variables, it was indicated that mortality from communicable diseases for males and females with elementary or less education was 2.82 and 2.73 times greater than that of those with third-level education, respectively. Therefore, in all the analyses, the magnitude of the association showed a clear gradient with educational level [28], emphasizing the important role of education level as a socioeconomic factor in the prognosis and outcome of infectious diseases. In this study, neurologic symptoms showed more prevalent among illiterate patients than among those without a high-school diploma. This result may be due to the fact that illiterate patients had a significantly higher mean age than those without a high-school diploma (75.7 ± 4.5 vs. 38.7 ± 14 years, *p* ≤ 0.001) (data not shown in the table). As the risk of developing neurological problems increases with age, it can affect the outcome of our study [29].

Our study has some limitations. As outpatients self-reported their clinical symptoms, there is a possibility of bias and inaccuracy in the expression of symptoms. Moreover, we did not assess body iodine status, which may have strengthened our study design. Our investigation included individuals with asymptomatic, pre-symptomatic/mild, and moderate COVID-19, and consequently, our findings cannot be generalized to those populations with a severe form of the disease.

## 5. Conclusions

Our data showed significant differences in the serum concentrations of T3 and T4 of outpatients infected with SARS-CoV2 and non-infected participants. We did not observe a correlation between baseline serum concentrations of TSH, T3, and T4 and the disease severity and the progression of its clinical symptoms. Moreover, we found that educational status and gender significantly affected the chance of neurological and pulmonary symptoms occurring during the study period. A higher prevalence of asthma and allergy comorbidities was found in the moderate positive-PCR group. Additionally, increased disease severity led to significant increases in mean respiratory and mean pulse rates. Our results may help develop potential therapies and public health strategies for the improvement of endocrine function and symptomatology of COVID-19 disease.

## Figures and Tables

**Figure 1 medicina-58-00891-f001:**
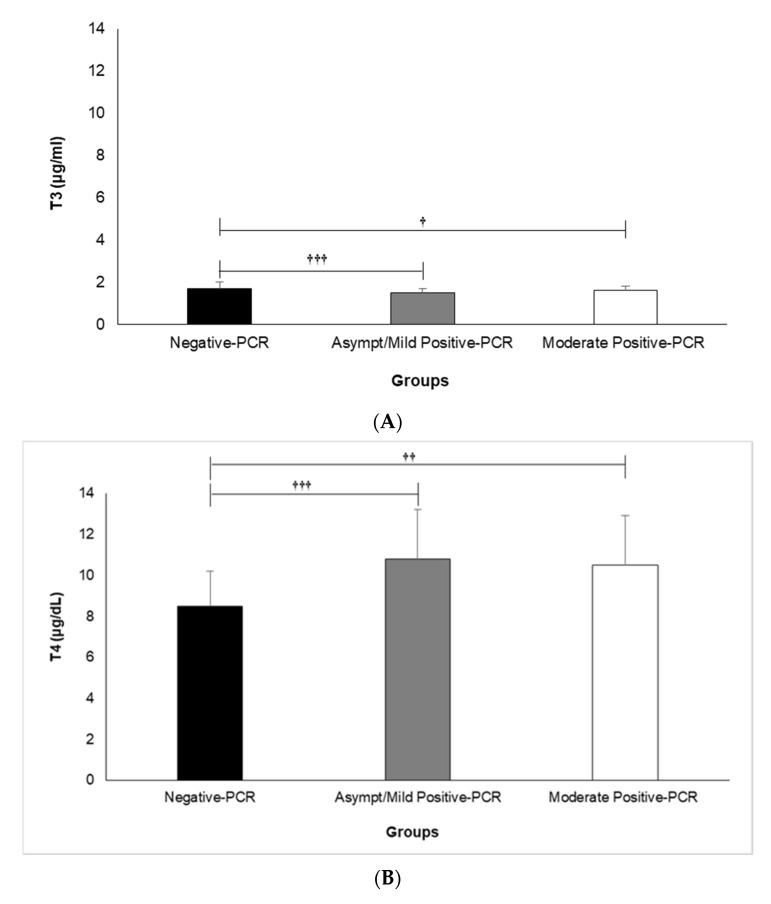
Comparison of Serum Levels of T3 (**A**), T4 (**B**), and TSH (**C**) among different clinical classifications of COVID-19 according to severity. Analysis of variance (ANOVA) was applied to analyze data. T3: triiodothyronine, T4: thyroxine, TSH: thyroid-stimulating hormone. † (*p* < 0.05), †† (*p* < 0.01), and ††† (*p* < 0.001).

**Table 1 medicina-58-00891-t001:** General characteristics, clinical presentations, and comorbidities of patients infected with SARS-CoV-2 and matched controls *.

	Groups		Negative PCR (n = 53)	Asymptomatic/Mild Positive PCR (n = 39)	Moderate Positive PCR (n = 14)	*p*-Value
Parameter						
General parameters		Age (year)	39.9 ± 13.4	42.5 ± 12.7	37.3 ± 13.2	0.41
		Sex				0.02
		Male	38.0 (70.4)	31.0 (79.5)	5.0 (38.5)	
		BMI ^+^ (Kg/m^2^)	27.5 ± 2.2	27.5 ± 4.0	25.7 ± 6.3	0.38
		Smoking	9.0 (16.7)	9.0 (23.1)	3.0 (23.1)	0.71
Marital status		Single	12.0 (22.2)	9.0 (23.1)	3.0 (23.1)	0.99
		Married	42.0 (77.8)	30.0 (76.9)	10.0 (76.9)	
Educational status		Illiterate	1.0 (1.9)	2.0 (5.3)	0.0 (0.0)	0.11
		Those without a high-school diploma	19.0 (35.8)	7.0 (18.4)	8.0 (61.5)	
		Diploma	10.0 (18.9)	10.0 (26.3)	3.0 (23.1)	
		College education	23.0 (43.4)	19.0 (50.0)	2.0 (15.4)	
Vital signs		Duration of infection (days)	0.0	6.6 ± 2.0	6.1 ± 1.3	0.04
		Respiratory rate (number/min)	13.0 ± 1.9	13.5 ± 1.1	15.9 ± 1.7	≤0.001
		Pulse rate (number/min)	87.2 ± 13.4	87.3 ± 17.0	100 ± 18.0	0.02
		SpO2 ^+^ (%)	97.4 ± 1.2	96.9 ± 1.3	97 ±1.6	0.14
Comorbidities		Hypertension	6.0 (11.1)	8.0 (20.5)	1.0 (7.7)	0.34
		Diabetes	5.0 (9.3)	3.0 (7.7)	2.0 (15.4)	0.71
		Obesity	22.0 (40.7)	10.0 (25.6)	2.0 (15.4)	0.12
		Malnutrition	1.0 (1.9)	0.0 (0.0)	0.0 (0.0)	0.61
		Cancer	1.0 (1.9)	1.0 (2.6)	0.0 (0.0)	0.84
		Liver disease	4.0 (7.4)	4.0 (10.3)	0.0 (0.0)	0.48
		Chronic pulmonary disease	1.0 (1.9)	1.0 (2.6)	0.0 (0.0)	0.84
		Chronic neurological disease	2.0 (3.7)	1.0 (2.6)	0.0 (0.0)	0.77
		Chronic hematologic disease	1.0 (1.9)	0.0 (0.0)	1.0 (7.7)	0.21
		Chronic renal disease	5.0 (9.3)	1.0 (2.6)	1.0 (7.7)	0.43
		Chronic heart disease	3.0 (5.7)	2.0 (5.1)	1.0 (7.7)	0.94
		AIDS/HIV ^+^	1.0 (1.9)	0.0 (0.0)	1.0 (7.7)	0.21
		Dementia	1.0 (1.9)	0.0 (0.0)	0.0 (0.0)	0.61
		Rheumatoid arthritis	1.0 (1.9)	1.0 (2.6)	0.0 (0.0)	0.84
		Asthma/Allergy	6.0 (11.1)	0.0 (0.0)	5.0 (38.5)	≤0.001
		Intestinal ulcer	0.0 (0.0)	1.0 (3.7)	0.0 (0.0)	0.32
		Other diseases	17.0 (31.5)	5.0 (12.8)	2.0 (15.4)	0.08
History		Has a history of blood transfusions or plasma products	0.0 (0.0)	0.0 (0.0)	0.0 (0.0)	-
		Hospitalization for the past two months due to respiratory problems	1.0 (1.9)	0.0 (0.0)	0.0 (0.0)	0.61
		History of contact with COVID-19 suspicious person	25.0 (46.3)	24.0 (61.5)	10.0 (76.9)	0.89
		The time interval between encountering a COVID-19 suspect person and visiting Imam Khomeini health center	20.4 ± 8.9	24.1 ± 11.5	25.4 ± 9.6	0.28
		Travel history in the last two months	14.0 (25.9)	8.0 (20.5)	2.0 (15.4)	0.66

* Independent sample *t*-test was applied to analyze quantitative data, and chi-square test was used to analyze categorical variables. The results are shown with mean ± standard deviation for quantitative and number (%) for categorical data. ^+^ BMI: body mass index, SpO_2_: oxygen saturation, AIDS/HIV: acquired immunodeficiency syndrome/human immunodeficiency virus.

**Table 2 medicina-58-00891-t002:** Odds ratio and 95% confidence interval (95% CI) estimated by GEE analysis with AR (1) model to determine the progression of the clinical symptoms and the associations with demographic and laboratory parameters among infected patients *.

Clinical Symptoms Category Parameters		General		Pulmonary		Gastrointestinal		Neurologic	
		OR (95% CI)	*p*-Value	OR (95% CI)	*p*-Value	OR (95% CI)	*p*-Value	OR (95% CI)	*p*-Value
General parameters	BMI	1.0 (0.9–1.1)	0.7	0.97 (0.9–1.0)	0.4	1.0 (0.9–1.1)	0.4	1.06 (1–1.1)	0.2
	Age	1.0 (0.9–1.1)	0.1	1.0 (1.0–1.1)	0.054	0.97 (0.9–1.0)	0.15	1.0 (0.9–1)	0.07
Gender	Male	0.9 (0.5–1.8)	0.9	0.7 (0.3–1.7)	0.5	1.2 (0.6–2.5)	0.53	0.4 (0.2–0.8)	0.007
	Female	1.0 (Ref)		1.0 (Ref)		1.0 (Ref)		1.0 (Ref)	
Education Status	College Education	1.7 (0.4–6.6)	0.5	11 (0.6–182.6)	0.1	0.9 (0.15–5.9)	0.95	0.4 (0.08–1.6)	0.18
	Diploma	1.1 (0.2–4.9)	0.9	9.4 (0.5–178.0)	0.14	0.9 (0.1–7.6)	0.9	0.3 (0.07–1.4)	0.14
	Those without a high-school diploma	1.2 (0.3–4.5)	0.8	20.9 (1.2–350.9)	0.03	0.8 (0.13–5.0)	0.82	0.1 (0.03–0.7)	0.017
	Illiterate	1.0 (Ref)		1.0 (Ref)		1.0 (Ref)		1.0 (Ref)	
Thyroid Function	T3	2.1 (0.2–22.4)	0.5	0.4 (0.02–7.6)	0.5	0.1 (0.013–1.7)	0.13	0.8 (0.07–9.5)	0.8
	T4	0.9 (0.8–1.0)	0.2	1.0 (0.8–1.2)	0.8	0.99 (0.9–1.1)	0.9	0.1 (1–1.1)	0.5
	TSH	1.0 (0.9–1.1)	0.4	1.0 (0.99–1.1)	0.1	1.0 (0.1–1.1)	0.2	1.06 (1–1.1)	0.4

* General estimation equation (GEE) was applied to analyze data. Odds of common clinical signs and symptoms of COVID-19 followed on days 1 to 28 of disease (95% CI). BMI: body mass index, TSH: thyroid-stimulating hormone, T3: triiodothyronine, T4: thyroxine.

## Data Availability

The data presented in this study are available on request from the corresponding author.

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
