# Peer review of "Serum Concentrations of Thyroid-Stimulating Hormone, Triiodothyronine, and Thyroxine in Outpatients Infected with SARS-CoV2 in Khuzestan Province, Iran: A Disease Clinical Course Approach"

_medicina, 2022, doi:10.3390/medicina58070891_

Round 1

Reviewer 1 Report

The manuscript highlights a critical problem related to COVID-19 which is the effect of COVID-19 infection on hypo and hyperthyroidism. It is important to understand that relationship which enhance our knowledge in dealing with COVID-19 pandemic and the infected patients. However, the following recommendations should be considered to enhance and improve the quality of the manuscript.

·      The number of patients used in this study is a little low (53 patients is not enough) and I was hopping to see higher patients number.

·      The authors continuously used the term (mild and moderate positive PCR). This is not appropriate! Instead, they should relay on the CT values for positive patients and use that as an indicator for mild/moderate positive patients.

·      This manuscript suffers from a serious issue in data presentation! The authors used only three tables to present their data and those tables are very crowded and not easy to follow at all! Further, the authors are not consistent in using the number of digits after the decimal palce (i.e. sometimes they used two numbers and sometimes they used one digit). Instead, they should present their data in a better format and present the significant difference between their reading very clearly.

Author Response

Subject: Medicina-1724316

Date: 2022.May.29

Manuscript ID: Medicina-1724316

Response to Reviewers

Dear editor in chief of the Medicina

Thank you for allowing us to submit a revised draft of the manuscript “Serum Concentrations of Thyroid-Stimulating Hormone, Triiodothyronine, and Thyroxine in Outpatients Infected with SARS-COV2 in Khuzestan Province, Iran: A Disease Clinical Course Approach” for publication in the Medicina. We appreciate the time and effort that you and the reviewers devoted to providing feedback on our manuscript and are thankful for the perceptive comments and valuable improvements to our manuscript. We have included all of the suggestions made by the reviewers and the editorial office. The changes are highlighted within the manuscript using green-colored text for reviewer 1 and using yellow-colored text for reviewer 2. Please see below, in red, for a point-by-point response to the comments and concerns.

We tried to ensure that our response letter adequately addressed each of the reviewer's and editorial office’s comments.

Sincerely,

Dr. Sahar Golabi, Ph.D. (Medical Physiology)

2022.May.29

Reviewers' Comments

Reviewer 1:

1-The number of patients used in this study is a little low (53 patients is not enough) and I was hoping to see a higher patient number.

Author response: The selection of the sample size was based on the opinion of the vital statistics consultant and based on the use of a similar article (the used reference is given in the text of the manuscript). Using the formula (which is also available in the text of the manuscript) the minimum sample size for healthy and outpatient groups was calculated as 38, which, taking into account the probabilities and losses, finally 53 individuals in each group were examined.

2- The authors continuously used the term (mild and moderate positive PCR). This is not appropriate! Instead, they should relay on the CT values for positive patients and use that as an indicator for mild/moderate positive patients.

Author response: Our study aimed to evaluate the outpatients of COVID-19 and their matched controls. That is, those who do not belong to the severe and/or critical grades. The outpatients fall into the asymptomatic/ mild and/or moderate categories of the disease. Since the disease of outpatients is not in the category of severe and/or critical, so the doctor did not prescribe CT for them, and for such individuals, only a PCR test was prescribed. The PCR test results, in addition to the clinical signs determined by the examination, as well as the fact that the participants' occupations were not at high risk, were our criteria for including individuals in the study and placing them in asymptomatic/ mild and/or moderate categories. People who did not have lung involvement entered our study.

3- This manuscript suffers from a serious issue in data presentation! The authors used only three tables to present their data and those tables are very crowded and not easy to follow at all! Further, the authors are not consistent in using the number of digits after the decimal place (i.e. sometimes they used two numbers and sometimes they used one digit). Instead, they should present their data in a better format and present the significant difference between their reading very clearly.

Author response: Thank you for pointing this out. We checked table 1. Based on your point, all numbers were expressed up to one decimal place and P-Values up to two digits. The expression form of all items was matched and the changed items were highlighted in green. The aberrations in Table 1 were footnoted below the table. Also, a column was added to the left of tables 1 and 2, which in addition to improving the presentation of the data, also helps to better read and understand the content presented in the tables.

Reviewer 2 Report

Work by Naghashpour et al. "Serum Concentrations of Thyroid-Stimulating Hormone, Triiodothyronine, and Thyroxine in Outpatients Infected with SARS-COV-2 in Khuzestan Province, Iran: A Disease Clinical Course Approach" is well written and presents an extremely interesting research topic.

My only comments about the work relate to its readability:

  1. In the title of the article, please change the name SARS-COV-2 to SARS-CoV2;
  2. The text of the abstract is correct, just a bit too long. I am asking you to consider whether it is not possible to shorten it a bit;
  3. Please enter the names T3, T4, TSH for the first time in the text, in the part introduction. And also pay attention whether all abbreviations used by you are expanded;
  4. In table 1, I would divide the described parameters into categories, which I would place on the side of the table in a separate column. This would allow a better presentation of the grouped results and improve their readability. For example, general category (age, sex, BMI, smoking), married status (single / married), comorbidities or parameters related to COVID-19 (all other parameters in the table). An example of such a table is attached in the file. Of course, please follow the journal's guidelines regarding the edition of the table itself (presence of individual dashes etc.)
  1. Please explain the abbreviations used in table 1 below;
  2. Please edit table 2 in the same way as table 1, it will improve the readability of the work and better reception of the results presented there;

Author Response

Subject: Medicina-1724316

Date: 2022.May.29

Manuscript ID: Medicina-1724316

Response to Reviewers

Dear editor in chief of the Medicina

Thank you for allowing us to submit a revised draft of the manuscript “Serum Concentrations of Thyroid-Stimulating Hormone, Triiodothyronine, and Thyroxine in Outpatients Infected with SARS-COV2 in Khuzestan Province, Iran: A Disease Clinical Course Approach” for publication in the Medicina. We appreciate the time and effort that you and the reviewers devoted to providing feedback on our manuscript and are thankful for the perceptive comments and valuable improvements to our manuscript. We have included all of the suggestions made by the reviewers and the editorial office. The changes are highlighted within the manuscript using green-colored text for reviewer 1 and using yellow-colored text for reviewer 2. Please see below, in red, for a point-by-point response to the comments and concerns.

We tried to ensure that our response letter adequately addressed each of the reviewer's and editorial office’s comments.

Sincerely,

Dr. Sahar Golabi, Ph.D. (Medical Physiology)

2022.May.29

Reviewer 2:

  • In the title of the article, please change the name SARS-COV-2 to SARS-CoV2;

Author response: Thank you for your attention. It is done.

2- The text of the abstract is correct, just a bit too long. I am asking you to consider whether it is not possible to shorten it a bit;

Author response: Thanks for your comment. It is done. The phrase " This study was performed on 53 outpatients with COVID-19 and 53 non-infected individuals (with negative PCR tests for SARS-CoV2) as controls" has been removed from the abstract.

3- Please enter the names T3, T4, TSH for the first time in the text, in the part introduction. And also pay attention whether all abbreviations used by you are expanded;

Author response: Thanks for your consideration. It is done. All abbreviations and their expansions were checked in the text. In all cases, when an item is first presented in the text, its expansion is stated and in front of it, its abbreviation is presented in parentheses.

  1. In table 2, I would divide the described parameters into categories, which I would place on the side of the table in a separate column. This would allow a better presentation of the grouped results and improve their readability. For example, general category (age, sex, BMI, smoking), married status (single / married), comorbidities or parameters related to COVID-19 (all other parameters in the table). An example of such a table is attached below:

Group

Parameters

Negative PCR (n = 53)

Asymptomatic/Mild Positive PCR (n = 39)

Moderate Positive PCR (n = 14)

p- value

General parameters

Age

Sex

BMI

Smoking

Married status

single

married

And so on…

Of course, please follow the journal's guidelines regarding the edition of the table itself (presence of individual dashes etc.)

Author response: Thank you for pointing this out. Table 1 was corrected according to the comment and journal's guidelines.

  • Please explain the abbreviations used in table 1 below;

Author response: Thank you for pointing this out. It is done.

  1. Please edit table 2 in the same way as table 1, it will improve the readability of the work and better reception of the results presented there;

Author response: Thank you for pointing this out. Table 2 was corrected according to the comment and journal's guidelines.

Round 2

Reviewer 1 Report

I am still not convinced with your justification for using "mild and moderate positive PCR and not building your argument based on the CT values.  Further, I strongly recommend you change your data presentation and combine tables with other format (i.e. bar chart, etc). Tables are still very confusing and not easy to follow. 

Author Response

Subject: Medicina-1724316

Date: 2022.Jun.03

Manuscript ID: Medicina-1724316

Response to Reviewer

Dear editor in chief of the Medicina

Thank you for allowing us to submit a revised draft of the manuscript “Serum Concentrations of Thyroid-Stimulating Hormone, Triiodothyronine, and Thyroxine in Outpatients Infected with SARS-COV2 in Khuzestan Province, Iran: A Disease Clinical Course Approach” for publication in the Medicina. We appreciate the time and effort that you and the reviewer devoted to providing feedback on our manuscript and are thankful for the perceptive comments and valuable improvements to our manuscript. We have included all of the suggestions made by the reviewer and the editorial office. The changes are highlighted within the manuscript using pink-colored text for reviewer 1 (round 2). Please see below, in red, for a point-by-point response to the comments and concerns.

We tried to ensure that our response letter adequately addressed each of the reviewer's and editorial office’s comments.

Sincerely,

Dr. Sahar Golabi, Ph.D. (Medical Physiology)

2022.Jun.03

Reviewers' Comments

Reviewer 2:

1- I am still not convinced with your justification for using "mild and moderate positive PCR and not building your argument based on the CT values.

Author response: As we said before, Our study aimed to evaluate the outpatients of COVID-19 and their matched controls. In such patients, pulmonary involvement does not occur and therefore the use of CT diagnostic tests is not routine and is not used. Also, to minimize the chance of false negatives, the control group did not include individuals with COVID-19-related symptoms (such as fever and shortness of breath), individuals exposed to patients with COVID-19, or individuals in high-risk occupations (such as health care personnel, bank employees, and public transport drivers) or from pre-reading offices. COVID-19 outpatients were categorized under the supervision of an infectious disease specialist (Dr. Sara Mobarak, Abadan University of Medical Sciences) and according to disease severity and prognosis using the Center for Disease Control and Prevention (CDC) criteria, which include the following. (1) Asymptomatic or pre-symptomatic infection: individuals who showed positive RT-PCR test results with no symptoms of COVID-19. (2) Mild illness: individuals with any of the symptoms of COVID-19 (e.g., fever, vomiting, nausea, diarrhea, muscle pain, headache, cough, sore throat, smell, taste disorders, and malaise) but no dyspnea, shortness of breath, or abnormal chest imaging. (3) Moderate illness: individuals who indicated evidence of lower respiratory disease during clinical assessment or imaging and oxygen saturation (SpO2) of ≥94% in room air at sea level [Çakırca et al., 2021]. None of the COVID-19 outpatients included in the study had a severe or critical illness.

The use of such a method to classify different grades of COVId-19 outpatients has been used as a standard and valid method in previous works (Golabi et al., 2022).

References:

  • Çakırca, G.; Çakırca, T.D.; Üstünel, M.; Torun, A.; Koyuncu, I. Thiol level and total oxidant/antioxidant status in patients with COVID-19 infection. Ir. J. Med. Sci. 2021, 1971, 1–6.
  • Golabi, S.;Ghasemi, Sh.;Adelipour, M.; Bagheri, R.; Suzuki, K.; Wong, A.; Seyedtabib, M.; Naghashpour, M. Oxidative Stress and Inflammatory Status in COVID-19 Outpatients: A Health Center-Based Analytical Cross-Sectional Study. Antioxidants. 2022, 11, 606, 1-17.

2- Further, I strongly recommend changing your data presentation and combining tables with another format (i.e. bar chart, etc). The tables are still very confusing and not easy to follow.

Author response: Thanks for your valuable comment. We changed the data presentation and combined tables with another format (figure). We replaced table 3 with Figure 1.

EDITORIAL OFFICE COMMENTS

  • Please check that all references are relevant to the contents of the manuscript.

Author response: It is done.

  • Any revisions to the manuscript should be marked up using the “Track Changes” function if you are using MS Word/LaTeX, such that any changes can be easily viewed by the editors and reviewers.

Author response: It is done

  • Please provide a cover letter to explain, point by point, the details of the revisions to the manuscript and your responses to the referees’ comments.

Author response: It is done.

Round 3

Reviewer 1 Report

The used figures are still inappropriate, the authors should be more creative in their selection of the figures styles and formate. Figures comprise massive error bars which is strange and weird.

Author Response

Subject: Medicina-1724316

Date: 2022.Jun.17

Manuscript ID: Medicina-1724316

Response to Reviewer

Dear editor in chief of the Medicina

Thank you for allowing us to submit a revised draft of the manuscript “Serum Concentrations of Thyroid-Stimulating Hormone, Triiodothyronine, and Thyroxine in Outpatients Infected with SARS-COV2 in Khuzestan Province, Iran: A Disease Clinical Course Approach” for publication in the Medicina. We appreciate the time and effort that you and the reviewer devoted to providing feedback on our manuscript and are thankful for the perceptive comments and valuable improvements to our manuscript. We have included all of the suggestions made by the reviewer. The changes are highlighted within the manuscript using blue-colored text Please see below, in red, for a point-by-point response to the comments and concerns.

We tried to ensure that our response letter adequately addressed each of the reviewer's comments.

Sincerely,

Dr. Sahar Golabi, Ph.D. (Medical Physiology)

2022.Jun.17

Reviewers' Comments

Comments and Suggestions for Authors

The used figures are still inappropriate, the authors should be more creative in their selection of the figures styles and format. Figures comprise massive error bars which is strange and weird.

Author response: Thanks for your consideration. We modified the presentation of figure 1 (A, B, C). Also, The error bars applied to the figures were based precisely on the results of the study, and the vertical axis was modified to better represent the figure.

This manuscript is a resubmission of an earlier submission. The following is a list of the peer review reports and author responses from that submission.